# An Overview of the Infectious Cycle of Bunyaviruses

**DOI:** 10.3390/v14102139

**Published:** 2022-09-28

**Authors:** Hani Boshra

**Affiliations:** Global Urgent and Advanced Research and Development (GUARD), 911 Rue Principale, Batiscan, QC G0X 1A0, Canada; hboshra@guardrx.org

**Keywords:** bunyavirus, structure, genomics, immune response, review

## Abstract

Bunyaviruses represent the largest group of RNA viruses and are the causative agent of a variety of febrile and hemorrhagic illnesses. Originally characterized as a single serotype in Africa, the number of described bunyaviruses now exceeds over 500, with its presence detected around the world. These predominantly tri-segmented, single-stranded RNA viruses are transmitted primarily through arthropod and rodent vectors and can infect a wide variety of animals and plants. Although encoding for a small number of proteins, these viruses can inflict potentially fatal disease outcomes and have even developed strategies to suppress the innate antiviral immune mechanisms of the infected host. This short review will attempt to provide an overall description of the order *Bunyavirales*, describing the mechanisms behind their infection, replication, and their evasion of the host immune response. Furthermore, the historical context of these viruses will be presented, starting from their original discovery almost 80 years ago to the most recent research pertaining to viral replication and host immune response.

## 1. The Order Bunyavirales

The *Bunyvirales* are an order composed of mostly single-stranded, tri-segmented RNA viruses that are transmitted by either rodent or arthropod vectors [1]. As of this publication, this order is made up of fourteen families (Arenaviridae, Cruliviridae, Discoviridae, Fimoviridae, Hantaviridae, Leishbuviridae, Mypoviridae, Nairoviridae, Peribunyaviridae, Phasmaviridae, Phenuiviridae, Tospoviridae, Tulsaviridae, and Wupedieviridae) (International Committee on Taxonomy of Viruses-ICTV). Until recently, the *Bunyavirales* were taxonomically classified as a family but were later re-classified as the only order of the class *Ellioviricetes* to commemorate the late virologist Richard M. Elliott’s (1954–2015) contributions to the field [2]. While Dr. Elliott broadened our understanding of many aspects of the molecular biology of the Bunyavirales, one of his greatest contributions involved the creation of a reverse genetics system, in which a combination of transfected plasmid cDNAs could be used to generate an infectious, replicative virus *de novo* [3,4]. Such systems of reverse genetics have been used to deduce factors affecting the virulence, infectivity, replication, protein synthesis, and viral assembly of a wide variety of bunyaviruses.

## 2. Virus Structure

Bunyaviruses are mostly composed of tri-segmented negative/ambisense RNA, referred to as the Large (L), Medium (M), and Small (S) segments (with the exception of the Arenaviridae, which are bi-segmented into L and S segments). It should be noted that all three segments have partially complementary 5′ and 3′ ends, providing a potential “panhandle” secondary structure which may interact with the virus’ RNA-dependent RNA polymerase (RdRp) [5,6]. The RNA segments are complexed to nucleoproteins (which are encoded by an open reading frame (ORF) located on the S-segment), forming ribonucleoprotein structures. These, in turn, are packaged into enveloped, spherical virions, varying between 80 and 120 nm [7]. It should also be noted that an RdRp is also packaged within the virus and is involved in both the transcription and genome replication of the virus [6]. This polymerase is encoded as the only ORF of the L-segment [6]. The outer structure of the virion is composed of dimeric glycoproteins, which are referred to as the N-terminal (Gn) and C-terminal (Gc) proteins [8], which are denoted based on their appearance on the M-segment. The *Bunyaviridae* genome may also encode for up to two non-structural proteins: the non-structural protein of the M (NSm) and S (NSs) [9], whose functions involve the inhibition of the host immune response [10,11,12,13] (see Figure 1).

## 3. Historical Perspective

The first recognized bunyavirus was isolated from *Aedes* mosquitos in western Uganda [14] and was named Bunyamwera virus (BUNV), based on the village from which it was found. It was this virus that ultimately served as the prototype and name giver for the family (subsequently reclassified as order). It was later found that serum from BUNV-infected individuals was able to cross-react with other viruses such as Cache Valley virus, Wyeomia virus, and Kairi virus [15], which ultimately led to the designation of a new antigenic group named the Bunyamwera serogroup [16]. Continued serological cross-reactivity studies then led to the organization of a “Bunyamwera supergroup,” composed of 11 serogroups encompassing 90 different viruses [17]. Further morphological studies found that viruses from this supergroup possessed similar morphologies [18], enabling the characterization of these viruses from a structural standpoint. As this Bunyamwera serological supergroup was found to exhibit conservation across serological and morphological lines, further investigation into their mode of infection and replication then followed.

Purification of these viruses through ultracentrifugation of infected cell lysates enabled the study of the genomic material used by this supergroup. Pioneering work by Petterson et al. [19,20] on Uukuniemi virus (UUKV) found that radiolabeled genomic material isolated from UUKV-infected BHK21 cells were susceptible to cleavage by RNAse, but not DNAse, thereby demonstrating that this virus was encoded by an RNA genome. It should also be noted from this study (through fractionation experiments) that certain proteins are associated with lipids, while another protein fraction was found associated with labeled RNA. These early results provided the first clues that these types of viruses were enveloped RNA viruses, with a lipid-anchored protein exterior and ribonucleoprotein complexes localized in the interior. This study also provided early electron micrographs demonstrating that UUKV had a spherical morphology, with a filamentous ribonucleoprotein complex, consistent with other viruses in this supergroup; despite the fact that UUKV was a tick-borne virus (unlike other members of the Bunyamwera, which were transmitted primarily through mosquitos), UUKV would ultimately be included with the supergroup.

Further electron microscopy studies on Uukuniemi virus RNPs found that the three RNA segments of the viruses were present in three length classes of 2.8, 1.7, and 0.7 microns. All three segments (referred to as the small, medium, and large segments) were circular and had the same protein-to-RNA ratio [20]. These experiments served as the foundation for more-detailed experiments describing bunyavirus transcription and replication. These results were consistent with subsequent studies on bunyavirus replication, using other viruses such as Bunyamwera (BUNV) [21], LaCrosse virus (LACV) [21,22] and Snowshoe Hare virus (SSHV) [23] showing a similar type of segmented RNA genomes.

Another example of early insights into bunyavirus replication involved the study of two other serologically indistinguishable viruses from the California serogroup, the Lumbo and Tahyna viruses, which ultimately provided some of the first clues into both the genomic structure and replication strategies of bunyaviruses [24]. Bouloy et al. found that these viruses had three distinct populations of RNA, which were later to be determined to correspond to the tri-segmented genome commonly associated with this family of viruses. Furthermore, structural studies of the extracts from these viruses showed a round virion (similar in shape to UUKV) composed of circular RNA ribonucleoprotein complexes [25,26].

Later work by Bouloy and Hannoun [26,27] on Lumbo viruses also showed that the purified virus possessed an enzyme that incorporated nucleotides into RNA; thus, confirming the existence of an RNA-dependent RNA-polymerase used by the virus for genomic replication. This was also confirmed in UUKV [28]. A subsequent study on Lumbo viruses found that labeled RNA extracted from the cytoplasm of infected BHK cells was found to express sense and antisense viral RNA, further suggesting the existence of an RNA-dependent RNA polymerase [25]. Furthermore, protein-labeling experiments of the cytoplasmic extract found that both the cytoplasmic and purified viral RNA co-sedimented as three different RNA populations, all associated with a 25 kDa protein. This suggested that all three segments associated with the 25 kDa protein form a ribonucleoprotein complex. Similar results were also confirmed through the study of the Uukuniemi virus [28], suggesting that this method of RNA replication may be consistent with all bunyaviruses.

Further studies of arthropod-transmitted viruses (also known as arboviruses) found another group of viruses also displayed similar morphological and biochemical similarities to the Bunyamwera supergroup; however, these viruses displayed no significant serological cross-reactivity. Ultimately, these viruses were grouped into a single taxonomic family called the *Bunyaviridae* based on their common morphologic and genomic structure [29]. As more viruses were classified into this group, more serological groups were characterized, and serogroups gave rise to new genera of bunyaviruses. Over the coming decades, with an increasing number of genera, the *Bunyaviridae* family became the largest family of RNA viruses [30] and was ultimately re-classified as the taxonomic order *Bunyavirales* in 2017 [31].

## 4. The Infectious Cycle of Bunyaviridae

Visualization of the entire replication cycle of BUNV has been performed by reverse genetics, with the addition of fluorescent proteins of the Glycoprotein C (Gc) [32]. When incubated with Vero cells, viral attachment and internalization (through endocytosis) were observed within 10 min. Once inside the cell, viral assembly and budding were observed at the Golgi apparatus, which was also found to undergo fragmentation prior to the generation of new glycoproteins.

Other studies attempting to visualize infection were performed by Wichgers et al., where fluorescent in situ hybridization was performed to visualize RVFV infection. While the synthesis of all three segments (S, M, L) was observed, it was found that the genomic packaging was highly heterogeneous, with nearly 40% of secreted virions lacking at least one of the three segments [33]. This result suggests that, in the case of RVFV, genome packaging is a non-selective process. Another example of packaging inefficiency was also demonstrated for the Crimean–Congo Hemorrhagic fever virus (CCHFV), where only a small fraction of virus particles possessed a complete genome [34].

## 5. Host Cell Receptors Used by Bunyaviruses for Cell Entry

### 5.1. DC-SIGN

Studies on severe fever with thrombocytopenia virus (SFTV) [35] have suggested a pH-dependent mechanism of viral entry, which is mediated through the host cellular protein Dendritic Cell-Specific Intracellular-3-Grabbing Non-integrin (DC-SIGN). These results are in agreement with previous studies on Rift Valley fever virus (RVFV) and Uukuniemi virus (UUKV) [36], where DC-SIGN was also implicated in virus attachment. Other experiments using pseudotyped VSV bearing the envelope glycoprotein of Crimean–Congo Hemorrhagic fever virus (CCHFV) found that DC-SIGN enhanced viral entry; thus, suggesting that this protein may play a role in the entry of a variety of bunyaviruses [37].

### 5.2. L-SIGN

Like DC-SIGN, L-SIGN (Liver/lymph node-specific ICAM-3-grabbing non-integrin) is a C-type lectin that has been implicated as a potential receptor in bunyavirus attachment. Sharing 77% sequence homology to DC-SIGN, studies using HeLa and Raji cells expressing L-SIGN were able to be infected by UUKV, RVFV, and TOSV [38].

### 5.3. Nucleolin

Immunoprecipitation studies using a recombinant portion of CCHFV glycoprotein C (GC), when incubated with cell extracts from Vero E6, 293T, and SW-13 cells, found that nucleolin strongly co-purified with GC [39]. As nucleolin has been previously associated with angiogenesis, Xiao et al. have hypothesized that the hemorrhagic effects associated with CCHFV infection may be linked to this protein [39].

### 5.4. Heparin Sulfate

Heparin sulfate proteoglycan (HSPG) is a glycoprotein that is expressed in all animal tissues [40]. It has been shown to serve a wide variety of functions, including blood coagulation and inflammation, as well as regulating cytokine function and cell adhesion [41]. HSPGs have also been previously shown to act as a substrate for virus attachment, as shown in studies involving herpes simplex virus (HSV) [42] and COVID-19 [43]. In bunyaviruses, HSPGs have been implicated as important attachment factors for Schmallenberg virus (SBV) and Akabane viruses (AKAV) [44], Toscana virus (TOSV) [45] and Rift Valley fever virus (RVFV) [46,47].

### 5.5. LRP1

Low-density lipoprotein receptor 1 (LRP1) is a membrane-bound receptor that plays a role in intracellular signaling and receptor-mediated endocytosis [48]. A recent study by Ganaie et al. identified proteins associated with RVFV infection by performing a genome-wide CRISPR/Cas9 screening of the murine microglial BV2 cell line [49]. By transducing cells with single guide RNAs (sgRNAs) targeting 20,000 unique genes, subsequent infection with virulent RVFV found multiple transformant cells that were resistant to RVFV cytopathogenesis. This work, as well as a parallel study using similar methods [50] identified LRP1, and Ganaie et al. also found that two proteins that facilitate the processing of LRP1 (i.e., RAP and Grp94) suppressed RVFV infection.

## 6. Uncoating and Viral Entry

Studies using different bunyaviruses have yielded varying mechanisms for viral entry. In the case of hantaviruses, several cellular proteins have been implicated in receptor-mediated attachment prior to entry [51]. These include β3 integrins [52], the complement receptor DAF/CD55 [53], the globular head of complement component C1a receptor (gC1qR) [54], and protocadherin-1 (PCDH1) [55]. The endocytic uptake of hantaviruses, however, remains controversial, as viral uptake appears to depend on the type of virus studied. In the case of Old World hantaviruses, they appear to be internalized via clathrin-mediated endocytosis. However, these results appear to differ from New World hantaviruses, where viral inhibition using Andes virus (ANDV) on primary lung epithelial cells suggests that viral entry can be either clathrin-dependent or independent [56]. Hantavirus entry has also been shown to be through acidic endosomes, ranging from pH 4.9–6.3 [57]. Upon pH-dependent fusion between the host cell-membrane and the enveloped virus, entry is concluded with the uncoating of the virus, releasing the nucleoprotein in the host cell cytoplasm.

## 7. Bunyavirus Viral Replication

As previously mentioned, bunyavirus genome viral replication and transcription both occur in the cytoplasm [58]. The replication is based on the synthesis of an intermediate, (+) sense “copy” cRNA (i.e., complementary to the (−) sense genomic vRNA) [59]. While the intermediate cRNA molecule is of identical length to the genomic vRNA, the viral mRNA can be observed as being 12-18 nucleotides longer. This difference in size is due to the “cap-snatching mechanism” previously shown to be used by other viruses (i.e., influenza) to enable mRNA translation in the host cell (as shown by Bouloy et al., when studying the 5′ mRNA segments of Germiston virus [60], where the endonuclease activity of the RdRp cleaves capped host mRNA) [59]. It should also be noted that the 3′end of the mRNA was found to be between 100–150 bp shorter than the intermediate cRNA.

TIe intermediate complementary RNA serves as a template genome replication, with the RdRp serving as a polymerase to produce the negatively single-stranded RNA segments used in subsequent viral packaging [61]. It should be noted that both viral transcription and viral replication are performed by the RdRp.

Although all three genome segments are replicated by the viral RdRp, the degree of replication of each segment is not equal. Studies on Uukuniemi virus (UUKV) found that the M-segment replicated at a higher degree (nearly five times) than the L-segment [62], while similar observations were also observed with LaCrosse virus (LACV) [63]. Further studies by Barr et al. using the 5′ and 3′ untranslated regions (flanked to a reporter gene) showed for BUNV that the M-segment displayed the highest degree of replication while the S-segment displayed the least (i.e., M > L > S) [64]. The authors hypothesized that these differences in RNA translation might be (at least partly) due to the degree of complementarity between the 5′ and 3′ UTRs (with both the M and L segments having significantly higher complementary nucleotides at their ends compared to the S-segment).

## 8. Factors Affecting Bunyavirus Replication-Viral Factors

### 8.1. NSm

As previously mentioned, the bunyavirus genome may also encode for up to two non-structural proteins (NSm and NSs); the non-structural protein encoded on the M-segment (NSm) has, to date, no clear function. A study of temperature-sensitive mutants of the Maguari orthobunyavirus found that deletions corresponding to the NSm open reading frame did not affect viral growth in cell culture [65]. However, studies of virus-like particles (VLPs) generated using reverse genetics of Bunyamwera virus found that M-segments lacking the N-terminal portion of NSm significantly impeded VLP production [66].

In the Akabane virus, recombinant mutants lacking the entire NSm coding region were not able to be rescued, suggesting that this gene is essential for viral replication [67]. In Rift Valley fever virus, the importance of NSm appears to be controversial; studies using RVFV deletion mutants (deficient in NSm) suggest that this protein is a virulence factor in mosquitos, with a lack of NSm inhibiting replication in midgut epithelial cells [68]. Other experiments using NSm-deleted recombinant RVFV found that this protein may play a role in inhibiting virus-induced apoptosis in Vero E6 cells [69]. However, other experiments using NSm-deleted recombinant RVFV found no significant changes in viral growth when assayed in Vero E6 cells [70,71]. Recombinant viruses deficient in NSm were also found to retain virulence and lethality when infected in rat animal models [72].

In other bunyaviruses such as Schmallenberg virus (SBV), recombinant viruses lacking NSm did not affect replication in interferon defective BHK-21 cells but showed reduced virulence interferon α/β knockout (IFNAR -/-) mice [73]. However, recombinant SBV lacking NSm did not demonstrate any decrease in virulence when assayed in cattle. NSm was also found to be dispensable in the Oropouche virus [74]. Therefore, the importance of NSm in bunyavirus replication appears to vary based on the type of virus studied.

### 8.2. NSs

NSs’ role in the inhibition of the host response was characterized well before its genetic characterization. Data published in 1957 by Plowright and Ferris described a change in the morphology of sheep kidney cells following RVFV infection, where “cytopathic changes are produced, including intranuclear inclusions, which are neutralized by immune sera” [75]. These observations were then expanded upon by Swanepoel and Blackburn [76], where indirect immunofluorescence from RVFV-infected serum, as well as electron microscopy, demonstrated the existence of filamentous structures formed in infected eosinophils. This led to the conclusion that RVFV was responsible for the production of a nuclear antigen. Ultimately, it was found that NSs, when expressed recombinantly, created these nuclear filaments [77]. Nucleolar targeting by NSs also been observed in other bunyaviruses, including SBV [78].

A clearer understanding of IFN-inhibition by NSs has since been elucidated, with NSs forming a complex with Sin3A-associated protein 30 (SAP30) and the transcription factor Ying Yang 1 (YY1), inducing DNA damage signaling in the host [12]. This damage induces the apoptotic pathway, resulting in programmed cell death. Furthermore, the NSs-SAP30-YY1 complex is also responsible for the inactivation of transcription factor IIH (TFIIH), which in turn inhibits the expression of IFN-β [79]. Encoded on the small segment of the bunyavirus genome, NSs is a non-structural protein associated with the suppression of IFN-*α*/*β*-induced transcription. Originally identified as an ORF encoded by non-overlapping genes in snowshoe hare virus (SSHV) [80] and LaCrosse virus (LACV) [81], this existence of NSs was found to be a relatively conserved component of bunyaviruses. Its function was ultimately found to be linked with the inhibition of the host IFN antiviral response [82]. Studies using reverse genetics to generate NSs-deleted Bunyamwera virus (BUNV) found that IFN induction by the deletion mutant activated NF-kB and was dependent on the IFN transcription factor IRF-3 [83]. Furthermore, in vitro studies using the IFN-beta promoter found that cells transfected NSs alone suppressed IFN-beta production. Similar results were also found when studying LACV [84]; moreover, this study additionally found that NSs served as a virulence factor only in mammalian cells, with no NSs-related inhibitory activity found in mosquito cells. It should also be noted that NSs of BUNV, LACV, and the related Schmallenberg (SBV) virus were all found to localize to the nucleus of infected cells to inhibit transcription initiation by RNA polymerase II and disrupt nucleoli [78,85,86,87]. The NSs of Rift Valley fever virus (RVFV), which also interferes with IFN induction and RNA polymerase II activity [79,88,89], additionally plays a role in inhibiting mRNA transport [90]. Further evidence of NSs as a virulence factor acting on the IFN system was confirmed when an NSs-deleted Rift Valley fever mutant (Clone-13) was found to be non-pathogenic pathogenicity in wild-type mice but highly pathogenic in IFN-deficient animals [91]. Studies on a variety of bunyaviruses have confirmed that NSs functions as a virulence factor, with recombinant NSs-deletions resulting in the attenuation of viruses normally found to be virulent in humans/animals. Other examples include Schmallenberg virus (SBV) [92], severe fever with thrombocytopenia virus (SFTSV) [93], Sandfly fever Sicilian virus (SFSV) [94,95], Toscana virus (TOSV) [96,97,98], Uukuniemi virus (UUKV) [99], and Oropouche virus (OROV) [74], to name a few. An overview of this section is summarized in Table 1.

## 9. Factors Affecting Bunyavirus Replication-Host Factors

### 9.1. Interferon-Stimulated Genes (ISGs)

As previously mentioned, a viral infection of the host cell can stimulate an immune response through the recognition of pathogen-associated molecular patterns (PAMPs). These patterns can come from the structural elements of the virus (i.e., the glycoprotein) or the genomic material that accumulates in the host cytoplasm following uncoating. In both cases, PAMPs are recognized by the host’s pattern recognition receptors (PRRs). Examples of PRRs include Toll-like receptors (TLRs), a retinoid-acid inducible gene I (RIG-I), and melanoma differentiation-associated protein 5 (MDA5). An extensive review of PRRs and PAMPs involved in the host restriction of bunyaviruses has been published by Lerolle et al. [100]. The recognition of PAMPs by PRRs results in the initiation of Interferon-stimulated genes (ISGs), which encode genes involved in the innate immune response. ISGs have been implicated in the host response to the *Bunyavirales* and include the following.

### 9.2. Interferon Regulatory Factors (IRFs)

Interferon regulatory factors (IRFs) are a family of transcription factors that play a variety of important roles in the immune response [101]. IRFs have been shown to be inducers of type I interferons, which can be activated through phosphorylation following the activation of PRRs. At least three different IRFs have been shown to be activated in the host cell following infection by bunyaviruses.

Interferon regulatory factor 1 (IRF1) is a nuclear factor that binds and activates the promoters of type I interferon genes [102]. IRF1 was originally identified in cell extracts following infection by Newcastle virus and has since been found to be induced following the infection of the host cell by a wide variety of viruses, including Hepatitis A virus (HAV), Hepatitis B (HBV), Dengue virus (DENV), as well as at least one member of the Bunyaviridae. Yan et al. [103] performed transcriptome analysis which demonstrated that THP-1 macrophages infected with SFTSV had increased levels of several transcription factors, including IRF1.

Interferon regulatory factor 3 (IRF3) has been shown to be activated (through phosphorylation) by virus infection or by double-stranded RNA [104]. Examples of IRF3 activation by bunyaviruses were demonstrated in BUNV, where NSs-deleted BUNV was found to be involved in IRF3-activated induction of IFN in vitro [83], thereby also implicating the role of NSs in suppressing this response. NSs was also shown to block IRF3 activation in LACV [87]. IRF3 was also implicated in the restriction of OROV infection in mice [105].

Like IRF3, Interferon regulatory factor 7 (IRF7) is activated through phosphorylation and plays a pivotal role in the induction of IFN gene transcription following viral infections [106]. As was the case with IRF3, IRF7 was also implicated in the restriction of OROV infection in mice [105]. Induction of IRF7 in multiple human cell lines has been demonstrated following infection by SFTSV [107], and its ability to induce IFN expression was inhibited by NSs [107].

### 9.3. The 2′5′OAS

2′,5′-oligoadenylate synthetase (2′5′OAS) is an enzyme that has been shown to degrade viral single-stranded RNA through the ATP-dependent activation of RNase L [108]. Furthermore, 2′5′ OAS has been shown to be an IFN-inducible enzyme that recognizes double-stranded RNA [108]. The effect of this enzyme on bunyavirus replication is not entirely clear. While studies using RNAse L-deficient mice did not restrict BUNV replication [109], studies using the Apeu virus suggested an increase in 2′5′ OAS following infection [110].

### 9.4. IFITM-1, IFITM-2 and IFITM-3

Interferon-induced transmembrane proteins (specifically IFITM-1, IFITM-2, and IFITM-3) have been previously shown to be induced following the infection of viruses, ranging from Influenza A [111] to Ebola [112] to HIV [113]. While the exact functions of IFITM-1 and IFITM-2 are not precisely known, IFITM-3 expression appears to affect membrane (and endosomal) fluidity, possibly preventing the emergence of viral genomes from the endosomal pathway [114]. In the *Bunyaviridae*, several viruses have been shown to be affected by the upregulation of these genes. Mudhasani et al. [115] demonstrated that Vero E6 cell lines overexpressing all three IFITM genes restricted the in vitro infection of LaCrosse virus (LACV), Andes virus (ANDV), Rift Valley fever virus (RVFV), and Crimean-Congo hemorrhagic fever virus (CCHFV). In the case of LACV and ANDV, overexpression of any of the three IFTMs significantly suppressed infection, while only IFITM-2 and IFITM-3 could suppress infection by RVFV. Curiously, none of the IFITMs were able to restrict the CCHFV infection.

### 9.5. ISG20

ISG20 is a 20 kDa protein that exhibits a 3′ to 5′ endonuclease activity predominantly directed towards single-stranded RNA [116]. This protein was previously found to induce the host IFN response following the infection of HeLa cells to vesicular stomatitis virus (VSV), influenza virus, and encephalomyocarditis (EMCV) [116]. It has also been found to be stimulated in response to Bunyamwera virus (BUNV) infection [117]. Feng et al. demonstrated that ISG20-expressing Vero cells significantly reduced BUNV replication; furthermore, using a minigenome system where virus-like particles were expressed with each of the three segments of the viral genome (cloned with a reporter gene), it was shown the ISG20-expressing HeLa cells could inhibit the expression of the reporter gene when expressed with either the L, M, or S segments.

### 9.6. PKR

Protein kinase R (PKR) is a ubiquitously expressed protein that is induced by cellular stress, including viral infection [118]. An example of its activity is its ability to be autophosphorylated and activated upon binding to double-stranded RNA. Once phosphorylated, PKR can then phosphorylate the eukaryotic translation initiation factor eIF2α, which in turn, inhibits all translation (thus, preventing the production of viral proteins). In the *Bunyaviridae*, Streitenfeld et al. demonstrated BUNV-activated PKR [109]. Keeping in mind that double-stranded RNA can activate the IFN-induction through toll-like receptors (TLRs), as well as directly interacting with PKR, Streitenfeld et al. attempted to see if NSs deleted BUNV affected both IFN induction, as well as PKR activation. Interestingly, while the NSs-deleted BUNV displayed reduced infection in cells pre-treated with IFN-α, there was no difference in virus replication between PKR knockout and wild-type MEF cells. This is different from the situation with phleboviruses like RVFV, SFSV, and TOSV, for whose NSs proteins, a specific anti-PKR activity was detected [95,119,120,121,122,123].

## 10. Conclusions

Since its original characterization as a serogroup of viruses associated with febrile illness over 80 years ago, considerable progress has been made in our understanding of bunyaviruses. From a phylogenetic standpoint, the number of identified bunyaviruses has exceeded 500, making them the largest group of RNA viruses. Over the span of decades, bunyaviruses were originally identified as a serogroup, to a family, to the current order *Bunyaviridae*. As the number of identified bunyaviruses increased, the pioneering work performed over the course of the last several decades has significantly increased our understanding of their structure, mode of infection, and replication. While much has been elucidated, a more definitive understanding of how bunyaviruses infect their hosts (both mammalian and insects), as well as a more detailed picture of the precise mode of virus replication and transcription, continues to be developed. Furthermore, our increased insights into the host immune response will enable the development of more potent antiviral therapies, which could be used to treat the hemorrhagic fevers caused by the more pathogenic viruses of this order, such as Hantaviruses, LASV, RVFV, and CCHFV.

## Figures and Tables

**Figure 1 viruses-14-02139-f001:**
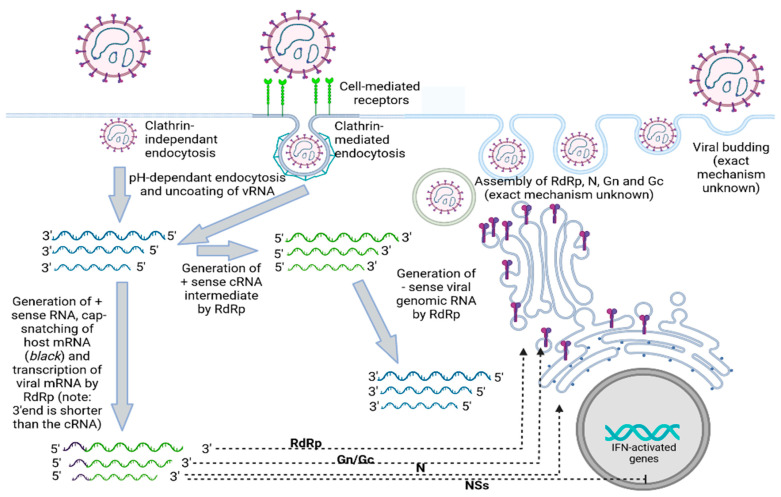
Schematic of our current understanding of the replication cycle of a typical bunyavirus. Bunyaviruses may be able to infect a cell either through receptor-mediated endocytosis (which may involve clathrin) or through clathrin-independent endocytosis. Upon viral entry, uncoating of the virion results in the presentation of multi-segmented (−) sense RNA in the host cytoplasm. An RNA-dependent RNA polymerase (RdRp) from the infectious particle enables the replication of the viral genome into a positive sense RNA intermediate. A “cap-snatching” mechanism enables the attachment of 5′host mRNA, resulting in translatable viral mRNA. Translation of the viral mRNA results in the synthesis of all the components of an infectious virus (i.e., glycoprotein N, glycoprotein C, nucleoprotein, and RdRp) as well as up to two non-structural proteins, NSs and NSm. NSs has been shown to be capable of inhibiting the transcription of interferon-induced genes. The (+) sense intermediate RNA can also serve as a template for genomic replication by the RdRp into (−) sense genomic RNA. The figure was generated using icons and templates from Biorender.com.

**Table 1 viruses-14-02139-t001:** Selected Viral/Host Factors That Affect Viral Infection and Replication.

Factor	Source (Virus/Host)	Bunyavirus Studied	Function	References
β3 integrins	-Host cells	-hantaviruses (NY-1 and SNV)	-viral attachment	[52]
DAF/CD55	-Host cells	-hantaviruses (Hantaan virus, PUUV)	-viral attachment	[53]
gC1qR	-Host cells	-hantaviruses (Hantaan virus)	-viral attachment	[54]
protocadherin-1	-Host cells	-hantaviruses (ANDV, SNV)	-viral attachment	[55]
DC-SIGN	-Host cells	-CCHFV	-viral attachment	[35,36]
L-SIGN	-Host cells	-RVFV, UUKV, LACV	-viral attachment	[38]
nucleolin	-Host cells	-CCHFV	-unclear	[39]
Heparin Sulfate Proteoglycan (HSPG)	-Host cells	-SBV, AKAV, TOSV, RVFV	-viral attachment	[40,41,42,43,44,45]
NSm	-Virus	-Maguari, AKAV, RVFV, SBV	-unclear (possible inhibition of viral-induced apoptosis)	[63,64,65,66,67,68,69,70,71,72]
NSs	-Virus	-LACV, BUNV, RVFV, SBV	-inhibits transcription of IFN-activated genes	[72,73,74,75,76,77,78,79,80,81,82,83,84,85,86,87,88,89,90,91,92,93,94,95,96,97]

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
