# Peer review of "An Overview of the Infectious Cycle of Bunyaviruses"

_viruses, 2022, doi:10.3390/v14102139_

Round 1
Reviewer 1 Report
The review of Boshra H. “An Overview Of The Infectious Cycle of Bunyaviruses” is a short review on general aspects of Bunyaviruses: historical perspective, virus structure, entry, replication and factors affecting Bunyavirus replication. It is a well-written and comprehensive short review that should be of interest to students and researchers looking for a rapid, general overview on Bunyaviruses.
However, some aspects need to be improved.
A chapter describing the mammalian hosts of Bunyaviruses is required that should list the main mammalian hosts of Bunyaviruses, the transmission routes, a brief description of the different symptoms and pathologies induced by these viruses, the main organs targeted, a brief epidemiological overview as well as a brief update on vaccines and treatments. This chapter could be introduced after “3_Historical Perspective” before “4_Uncoating and Viral Entry”.
Chapter “8_The Infectious Cycle of Bunyaviruses” should be placed after chapter “6_Bunyavirus Viral Replication”.
The chapter “7_Factors Affecting Bunyavirus Replication” should be divided in two parts “Viral factors affecting replication” and “Host factors affecting replication” with Interferon Stimulated Genes (ISGs) included in “Host Factors affecting replication”.
The role of the viral non-structural NSs protein in regulating (or not regulating) viral replication in insect and mammalian cells needs to be added.
Even though the role of the viral NSs protein in regulating the host transcriptional capacity and type I IFN response is abundantly described, the information on the capacity of the NSs protein to form nuclear filamentous structures is lacking and needs to be discussed with respect to the capacity of the NSs protein to affect the host gene expression, cell and nuclear structure. This aspect of the viral NSs protein that has been described mainly in the case of RVFV should also be discussed with respect to other Bunyaviruses.
Concerning chapter “9_Interferon Stimulated Genes”:
-PRRs involved in the recognition of Bunyaviruses need to be indicated.
-The logic behind the choice of the ISGs discussed in the review with respect to the infectious cycle of Bunyaviruses is not clear. Why choosing IRF1 and not IRF3 or IRF7 that are the main regulators of type I IFNs in most cell types ? Why choosing IFITM1-3 and avoid OAS family of genes that constitute one of the major families of ISGs and play an essential role in regulating viral replication ?
Concerning References: of the 110 references cited, only 15 were published after 2018. Authors should make an effort to cite more recent work.
Author Response
The review of Boshra H. “An Overview Of The Infectious Cycle of Bunyaviruses” is a short review on general aspects of Bunyaviruses: historical perspective, virus structure, entry, replication and factors affecting Bunyavirus replication. It is a well-written and comprehensive short review that should be of interest to students and researchers looking for a rapid, general overview on Bunyaviruses.
However, some aspects need to be improved.
Response: The Author would like to thank Reviewer #1 for his/her kind words, and has made every attempt to address his/her concerns. The Author believes that the following suggestions have greatly improved the manuscript; and it is the hope of the Author that Reviewer #1 feels the same way. Please note that all changes to the manuscript have been sent to the editor in a Word document, a can be seen under “Track Changes”.
A chapter describing the mammalian hosts of Bunyaviruses is required that should list the main mammalian hosts of Bunyaviruses, the transmission routes, a brief description of the different symptoms and pathologies induced by these viruses, the main organs targeted, a brief epidemiological overview as well as a brief update on vaccines and treatments. This chapter could be introduced after “3_Historical Perspective” before “4_Uncoating and Viral Entry”.
Response: The Author thanks Reviewer #1 for his/her comments. While the Author finds all of these topics incredibly important (i.e. Epidemiology, Pathology, Hosts and Vaccine/Treatments), the Author feels as though all these topics would be better served as an entirely separate review. Keep in mind that the Author was approached by “Viruses” to write a short review pertaining to replication of bunyaviruses. To give Reviewer #1 an idea, the Author previously wrote a review in the Journal of Virology, solely on vaccine strategies against Rift Valley fever virus (Boshra et al, JVI, 2011 Jul;85(13):6098-105 doi: 10.1128/JVI.02641-10); and despite working on such a very narrow topic, it provided enough information to be published by itself. On the other hand, what Reviewer #1 is proposing is a very good idea. If the Editors decide to publish another Special Edition dedicated to the Bunyavirales (and assuming that they would like to ask the Author for another submission), a review on the Epidemiology, Pathology, Hosts and Vaccine/Treatments of Bunyaviruses would definitely be something the Author would be willing to write.
Chapter “8_The Infectious Cycle of Bunyaviruses” should be placed after chapter “6_Bunyavirus Viral Replication”.
Response: The Author agrees with Reviewer #1, in that the order of chapters appears to be incorrect. Reviewer #2 also had issues with the ordering of the chapters, suggesting that, “For example, Part 8 should be put in the very beginning position to start all review instead of the ending part. Also, part 4-Uncoating and Viral Entry should be changed with Part 5 Host cell receptor for attachment. First step is about virus attachment”. As some of the suggestions made by Reviewers #1 and #2 conflict with each other, the Author would like to propose a compromise for Reviewers #1 and #2. The following changes have been made with respect to the order of chapters:
Section 8 (The Infectious Cycle of the Bunyaviridae) is now Section 4
Section 5 (Host Cell Receptors Used By Bunyaviruses For Cell Entry) is still Section 5
Section 4 (Uncoating and Viral Entry) is now Section 6
Section 7 (Bunyavirus Viral Replication) is still Section 7
Factors Affecting Replication—Viral Factors) is now Section 8
Section 9 (ISGs) is still Section 9, but changed to Factors Affecting Replication—Host Factors
The Authors thanks Reviewer #1 for the suggestion
The chapter “7_Factors Affecting Bunyavirus Replication” should be divided in two parts “Viral factors affecting replication” and “Host factors affecting replication” with Interferon Stimulated Genes (ISGs) included in “Host Factors affecting replication”.
Response: The Author agrees with this suggestion, and has separated “Factors Affecting Bunyavirus Resplication” into a “Host” and “Viral” section. Keep in mind, that as Reviewer #2 suggested that Chapter 7 be merged into Chapter 6, the Author has now made “Viral” into Chapter 8 and “Host” into Chapter 9 in order to satisfy both Reviewers.
The role of the viral non-structural NSs protein in regulating (or not regulating) viral replication in insect and mammalian cells needs to be added.
Even though the role of the viral NSs protein in regulating the host transcriptional capacity and type I IFN response is abundantly described, the information on the capacity of the NSs protein to form nuclear filamentous structures is lacking and needs to be discussed with respect to the capacity of the NSs protein to affect the host gene expression, cell and nuclear structure. This aspect of the viral NSs protein that has been described mainly in the case of RVFV should also be discussed with respect to other Bunyaviruses.
Response: The Author has added the following to the section on Nss
NSs’ role in the inhibition of the host response was characterized well before its genetic characterization. Data published in 1957 by Plowright and Ferris described a change in the morphology of sheep kidney cells following RVFV infection, where “cytopathic changes are produced, including intranuclear inclusions, which are neutralized by immune sera” [94]. These observations were then expanded upon by Swanepoel and Blackburn [95], where indirect immunofluorescence from RVFV-infected serum, as well as electron microscopy, demonstrated the existence of filamentous structures formed in infected eosinophils. This led to the conclusion that RVFV was responsible for the production of a nuclear antigen. Ultimately, it was found that NSs, when expressed recombinantly, created these nuclear filaments [96]. Nucleolar targeting by NSs also been observed in other bunyaviruses, including SBV [77].
A clearer understanding of IFN-inhibition by NSs has since been elucidated, with NSs forming a complex with Sin3A-associated protein 30 (SAP30) and the transcription factor Ying Yang 1 (YY1), inducing DNA damage signaling in the host [12]. This damage induces the apoptotic pathway, resulting in programmed cell death. Furthermore, the NSs-SAP30-YY1 complex is also responsible for the inactivation of transcription factor IIH (TFIIH), which in turn inhibits the expression of IFN-β [82]
Concerning chapter “9_Interferon Stimulated Genes”:
-PRRs involved in the recognition of Bunyaviruses need to be indicated.
Response: Reviewer #1 is correct in suggesting that more PRR’s should be indicated. Although the Author already mentioned RIG-I, the author has also added MDA5 as well. The author has also suggested reading a review from LeRolle et al (2021) for a more in-depth review of host-cell restriction factors.
-The logic behind the choice of the ISGs discussed in the review with respect to the infectious cycle of Bunyaviruses is not clear. Why choosing IRF1 and not IRF3 or IRF7 that are the main regulators of type I IFNs in most cell types ? Why choosing IFITM1-3 and avoid OAS family of genes that constitute one of the major families of ISGs and play an essential role in regulating viral replication ?
Response: The Author agrees with Reviewer #1, and has added sections describing OAS, IRF3 and IRF7. The Author thanks Reviewer #1 for the suggestion.
Concerning References: of the 110 references cited, only 15 were published after 2018. Authors should make an effort to cite more recent work.
Response: The Author appreciates Reviewer #1’s comments. It is true that most of the citations were before 2018. This was mainly due to the Historical Perspective, which was a main focus of this paper. The Author tried to do something a little different from other bunyavirus reviews, and tried to discuss the major discoveries over the last 80 years in (mostly) chronological order. Therefore a considerable number of the original, older work was cited. Therefore, when designing a review this way, there will ultimately be more cited from the first 76 years than the last 4. Nevertheless, the Author has added some newer references (thanks to both Reviewer #1 and Reviewer #2 for their suggestions).
The Author has also added a couple of more receptors for bunyavirus attachment (L-SIGN and Nucleolin), in order to make the review more extensive, and to add a few more recent publications.
L-SIGN
Like DC-SIGN, L-SIGN (Liver/lymph node-specific ICAM-3-grabbing non-integrin) is a C-type lectin that has been implicated as a potential receptor in bunyavirus attachment. Sharing 77% sequence homology to DC-SIGN, studies using HeLa and Raji cells expressing L-SIGN were able to be infected by UUKV, RVFV and TOSV (Leger 2016).
Nucleolin
Immunoprecipitation studies using a recombinant portion of CCHFV glycoprotein C (GC), when incubated with cell extracts from Vero E6, 293T and SW-13 cells found that nucleolin strongly co-purified with GC (Xiao 2011). As nucleolin has been previously associated with angiogenesis, Xiao et al, have hypotheszed that the hemorrhagic effects associated with CCHFV infection may be linked to this protein (Xiao 2011).

Reviewer 2 Report
The author gave a very broad review on bunyaviruses. A major part of this review is about the historical context of these viruses with almost 80 years time.
First, Part 3 Historical Perspective- could be found in many of textbook about virology. I would like to suggest they can be minimized. Or most important research progress for each step of virus life cycle could be put i each step.
Second suggestion is that it can be more focused in the key steps in life cycle of bunyavirus, for example, attachment, cell entry, uncoating, replication, package and release. There are so many host factors and viral proteins invovled in each of these steps.
Meanwhile, these contents should be presented in a logic ways and in different layers instead of mix and key steps lose. For example, Part 8 should be put in the very beginning position to start all review instead of the ending part. Also, part 4-Uncoating and Viral Entry should be changed with Part 5 Host cell receptor for attachment. First step is about virus attachment.
Again, 7-Factors Affecting Bunyavirus Replication can be grouped together with Virus replication in a same layer.
At last but not least, why it should be a single part of part-9 the Interferon-stimulated genes (ISGs). It is a kind of host immune respons upon virus infection. There are lots of new progress about host immune-virus interaction in recent years.
Author Response
The Author gave a very broad review on bunyaviruses. A major part of this review is about the historical context of these viruses with almost 80 years time.
Response: The Author would like to thank Reviewer #2 for going through this manuscript. The Author has attempted to address each concern that has been raised, and believes that the suggestions provided have significantly improved the manuscript. Please note that all changes to the manuscript have been sent to the editor in a Word document, a can be seen under “Track Changes”.
First, Part 3 Historical Perspective- could be found in many of textbook about virology. I would like to suggest they can be minimized. Or most important research progress for each step of virus life cycle could be put i each step.
Response: The Author appreciates the criticism put forward by Reviewer #2. However, the Author is not sure about the idea that “many textbooks” provide the Historical Perspective of bunyavirus research (perhaps apart from the textbook “The Bunyaviridae” by Richard Elliott). The Author believes that this approach is rather novel for a short review (and is considerably easier to read than the 300+ pages of the aforementioned textbook). It was the Author’s intention to provide the general public (or newcomers to the field) with a greater appreciation of the pioneering work that led to our current understanding of these viruses.
Second suggestion is that it can be more focused in the key steps in life cycle of bunyavirus, for example, attachment, cell entry, uncoating, replication, package and release. There are so many host factors and viral proteins invovled in each of these steps.
Meanwhile, these contents should be presented in a logic ways and in different layers instead of mix and key steps lose. For example, Part 8 should be put in the very beginning position to start all review instead of the ending part. Also, part 4-Uncoating and Viral Entry should be changed with Part 5 Host cell receptor for attachment. First step is about virus attachment.
Response: Reviewer #2 is correct in pointing out that some of the sections covering infection/replication are out of sequence; however Reviewer #1 also had concerns about the ordering of chapters, suggesting that the “Bunyavirus Viral Replication chapter be divided in “Host” and “Viral” sections. Therefore, the Author has proposed the following compromise that will hopefully satisfy Reviwers #1 and #2.
Section 8 (The Infectious Cycle of the Bunyaviridae) is now Section 4
Section 5 (Host Cell Receptors Used By Bunyaviruses For Cell Entry) is still Section 5
Section 4 (Uncoating and Viral Entry) is now Section 6
Section 7 (Bunyavirus Viral Replication) is still Section 7
Factors Affecting Replication—Viral Factors) is now Section 8
Section 9 (ISGs) is still Section 9, but changed to Factors Affecting Replication—Host Factors
The Authors thanks Reviewer #2 for the suggestion
Again, 7-Factors Affecting Bunyavirus Replication can be grouped together with Virus replication in a same layer.
Response: The Author appreciates Reviewer #2’s suggestion; however Reviewer #1 suggested that the factors associated with viruses be presented in a separate chapter, while factors associated with the host response be presented in another chapter. In the end, while incorporating Reviewer #2’s re-ordering of the chapters, the Author split Chapter 7 into Viral Factors (Chapter 8) and Host Factors (Chapter 9). Hopefully, this could serve as a “compromise” between Reviewer #1 and Reviewer #2.
At last but not least, why it should be a single part of part-9 the Interferon-stimulated genes (ISGs). It is a kind of host immune respons upon virus infection. There are lots of new progress about host immune-virus interaction in recent years.
Response: Reviewer #2 is correct in that a lot of work on ISGs have been performed in recent years. While the Author only wanted to provide an extremely brief over of ISGs (an in-depth review of ISGs could easily be another review paper—if not an textbook chapter), the Author only has limited space in such a short review. However, the Author has added a discussion of IRF3 and IRF 7 (instead of just IRF1), as well as a discussion of the OAS/RNse L response. The Author would like to thank Reviewer #2 for this suggestion.
